# Lip to Speech Synthesis with Visual Context Attentional GAN

**Minsu Kim, Joanna Hong, Yong Man Ro**[*]
Image and Video Systems Lab
KAIST
{ms.k, joanna2587, ymro}@kaist.ac.kr

## Abstract

In this paper, we propose a novel lip-to-speech generative adversarial network, Visual Context Attentional GAN (VCA-GAN), which can jointly model local and global lip movements during speech synthesis. Specifically, the proposed VCA-GAN synthesizes the speech from local lip visual features by finding a mapping function of viseme-to-phoneme, while global visual context is embedded into the intermediate layers of the generator to clarify the ambiguity in the mapping induced by homophene. To achieve this, a visual context attention module is proposed where it encodes global representations from the local visual features, and provides the desired global visual context corresponding to the given coarse speech representation to the generator through audio-visual attention. In addition to the explicit modelling of local and global visual representations, synchronization learning is introduced as a form of contrastive learning that guides the generator to synthesize a speech in sync with the given input lip movements. Extensive experiments demonstrate that the proposed VCA-GAN outperforms existing state-of-the-art and is able to effectively synthesize the speech from multi-speaker that has been barely handled in the previous works.

## 1 Introduction

Lip to speech synthesis (Lip2Speech) is to predict an audio speech by watching a silent talking face video. While conventional visual speech recognition tasks require human annotations (*i.e.*, text), Lip2Speech does not require additional annotations. Thus, it has drawn big attention as another form of lip reading. However, due to the ambiguity of homophenes that have similar lip movements and the voice characteristics varying from different identities, it is still considered as a challenging problem.

Basically, synthesizing a speech from a silent lip movement video can be viewed as finding a mapping function of visemes into corresponding phonemes. However, only watching short clip-level (*i.e.*, local) lip movements could be challenging to distinguish the homophenes. Thus, global-level lip movements containing the visual context, hints for ambiguity of viseme-to-phoneme mapping, should also be considered along with local-level lip movements. Early deep learning-based works [1, 2, 3] predict each auditory feature (*e.g.*, LPC, mel-spectrogram, spectrogram) within a short video clip and extend the prediction to the entire speech by sliding a window over the whole video sequences. As they operate with clip-level videos of fixed length, they could fail on capturing the global context of the spoken speech. A recent work [4] brings Sequence-to-Sequence (Seq2Seq) architecture [5, 6] that predicts the auditory feature conditioned on both the encoded visual context and the previous prediction and shows a promising performance. However, since they do not explicitly consider local visual features, they may produce out-of-sync speech to the input video. Moreover, due to the sequential nature of the Seq2Seq architecture, the method demands heavy inference time. Since the

---

[*]Corresponding author.

output audio sequence length is determined when the input video is given for the Lip2Speech task, the time costs can be reduced by adopting different architectures that could predict the speech with one forward step, instead of using a Seq2Seq model which is originally designed to handle input and output with different varying sequence lengths. Lastly, all the above methods focus on handling speech synthesis of constrained speakers (*i.e.*, 1 to 4 speakers), so they could fail to properly handle diverse speakers with one trained model.

In this paper, we design a novel deep architecture, namely Visual Context Attentional GAN (VCA-GAN), that jointly models the local and global visual representations to synthesize accurate speech from silent talking face video. Concretely, the proposed VCA-GAN synthesizes the speech (*i.e.*, mel-spectrogram) based on the local visual features by finding a mapping function of viseme-to-phoneme, while the global visual context assists the generator for clarifying the ambiguity of the mapping. To this end, a visual context attention module is proposed where it extracts the global visual features from the local visual features and provides the global visual context to the generator. It is applied to the generator in multi-scale scheme so that the generator can refine the speech representation from coarse- to fine-level by jointly modelling both the local and the global visual context. Moreover, to guarantee the generated speech to be synced with the input lip movements, synchronization learning is performed that gives feedback to the generator whether the synthesized speech is synchronized or not with the input lip movement. The effectiveness of the proposed framework is evaluated on three public benchmark databases, GRID [7], TCD-TIMIT[8], and LRW[9] in both constrained-speaker setting and multi-speaker setting.

The major contributions of this paper are as follows, 1) To the best of our knowledge, this is the first work to explicitly model the local and global lip movements for synthesizing detailed and accurate speech from silent talking face video. 2) We consider a mel-spectrogram as an image and solve the Lip2Speech problem efficiently using video-to-image translation. 3) This paper introduces synchronization learning which guides the generated mel-spectrogram to be in sync with the input lip video. 4) We show the proposed VCA-GAN can synthesize speech from diverse speakers without the prior knowledge of speaker information such as speaker embeddings.

## 2   Related Work

**Lip to Speech Synthesis.** There have been a number of researches and interests in visual-to-speech generation. Ephrat *et al.*[1] utilized CNN to predict acoustic features from silent talking videos. Then, they augmented the model to two-tower CNN-based encoder-decoder [2] where each tower encodes raw frames and optical flows, respectively. Akbari *et al.*[3] employed a deep autoencoder for reconstructing the speech features from the visual features encoded by a lip-reading network. Vougioukas *et al.*[10] directly synthesized the raw waveform from the video by using 1D GAN. Prajwal *et al.*[4] focused on learning lip sequence to speech mapping for a single speaker in an unconstrained, large vocabulary setting using a stack of 3D convolution and Seq2Seq architecture. Yadav *et al.*[11] used stochastic modelling approach with variational autoencoder. Michelsanti *et al.*[12] predicted vocoder features of [13] and synthesized speech using the vocoder. Different from the previous works, our approach explicitly models the local visual feature and global visual context to synthesize accurate speech. Moreover, we try to synthesize the speech from multi-speaker which has rarely been handled in the past.

**Visual Speech Recognition (VSR).** Parallel to the development of Lip2Speech, Visual Speech Recognition (VSR) have achieved a great advancement [14, 15, 16, 17, 18, 19]. Slightly different from the Lip2Speech, VSR identifies spoken speech into text by watching a silent talking face video. Several works have recently showed state-of-the-art performances in word- and sentence-level classifications. Chung *et al.*[9] proposed a large-scale audio-visual dataset and set a baseline model for word-level VSR. Stafylakis *et al.*[20] proposed an architecture that is combined of residual network and LSTM, which became a popular architecture for word-level lip reading. Martinez *et al.*[21] replaced the RNN-based backend with Temporal Convolutional Network (TCN). Kim *et al.*[19, 22] proposed to utilize audio modal knowledge through memory network without audio inputs during inference for lip reading. Assael *et al.*[23] achieved end-to-end sentence-level lip reading network by adopting the CTC loss [24]. Different from the VSR methods, the Lip2Speech task does not require human annotations, thus is drawing big attention with its practical aspects.

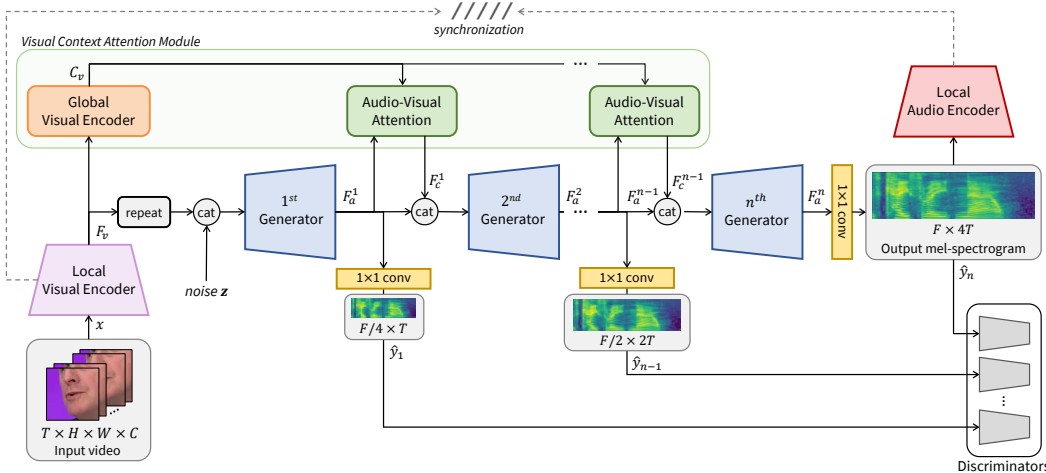

Figure 1: Overview of the VCA-GAN. Global visual context is provided through proposed visual context attention module to the generators to refine the speech representation from low- to high-resolution.

**Attention Mechanism.** The attention mechanism has affected many research fields, such as image captioning [25, 26, 27], machine translation [28, 29], and speech recognition [30, 31, 32]. It can effectively focus on relative information and reduce the interference from less significant one. There have been several works that incorporate attention mechanism in GAN. Xu *et al.*[25] proposed a cross modal attention model to guide the generator to focus on different words when generating different image sub-regions. Qiao *et al.*[27] further developed it by proposing a global-local collaborative attentive module to leverage both local word attention and global sentence attention and to enhance the diversity and semantic consistency of the generated images. Li *et al.*[26] introduced channelwise attention-driven generator that can disentangle different visual attributes, considering the most relevant channels in the visual features to be fully exploited. In this paper, we design a cross-modal attention module working with video and audio modalities for context modelling during speech synthesis. By bringing the global visual context through the proposed visual context attention module, speech of high intelligibility can be synthesized.

## 3 Proposed Method

Since an audio speech and lip movements in a single video are supposed to be aligned in time, the speech can be synthesized to have the same duration as the input silent video. Let $x \in \mathbb{R}^{T \times H \times W \times C}$ be a lip video with $T$ frames, height of $H$, width of $W$, and channel size of $C$. Then, our objective is to find a generative model that synthesizes a speech $y \in \mathbb{R}^{F \times 4T}$, where $y$ is a target mel-spectrogram with $F$ mel-spectral dimension and frame length of $4T$. The frame length of mel-spectrogram is designed to be 4 times longer than that of video by adjusting the hop length during Short-Time Fourier Transform (STFT). To generate elaborate speech representations, the proposed VCA-GAN (Fig.1) refines the viseme-to-phoneme mapping with the global visual context obtained from a visual context attention module, and learns to produce a synchronized speech with given lip movements. Please note that we treat the mel-spectrogram as an image and train the model with 2D GAN [33, 34].

### 3.1 Visual context attentional GAN

Considering the entire context from the input lip movements, namely global visual context, can provide additional information that alleviates the ambiguity of homophenes besides the accurate temporal alignment of local visual representations. To achieve this, the generator synthesizes the speech from the local visual features while the global visual context is jointly considered at the intermediate layers of generator through the visual context attention module. Firstly, a local visual encoder $\phi_v$ encodes the video $x$ into local visual features $F_v = \{f_v^1, f_v^2, \cdots, f_v^T\} \in \mathbb{R}^{T \times D}$, where $D$ is the dimension of embedding. The local visual encoder $\phi_v$ is composed of combination of 3D and 2D convolutions. Due to the locality of the convolution operator, each local visual

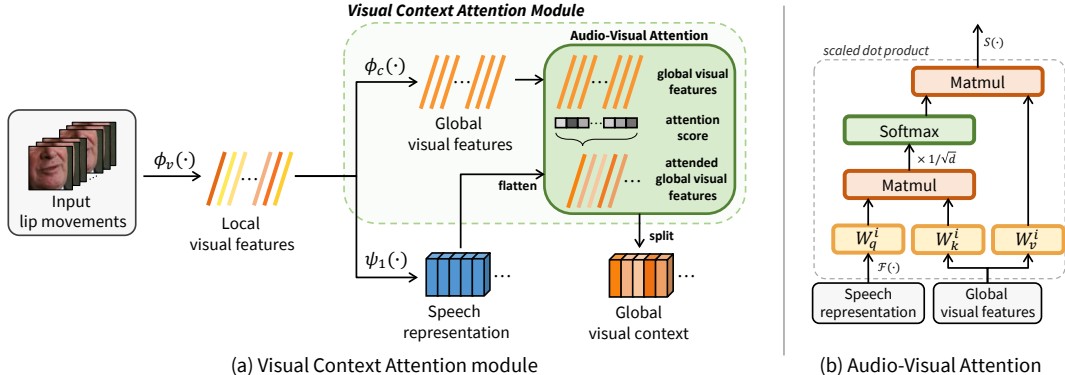

(a) Visual Context Attention module

(b) Audio-Visual Attention

Figure 2: Illustration of visual context modelling through the proposed visual context attention module. (a) visual context module, and (b) audio-visual attention in detail.

feature $f_v^t$ contains short-clip level (*i.e.*, local) lip movements embedded through the 3D convolution. From the local visual features $F_v$, $n$ generators $(\psi_1, \psi_2, \ldots, \psi_n)$ gradually generate and refine the speech representation from low to high resolution. The first generator $\psi_1$ synthesizes coarse speech representation $F_a^1$ with the following equation,

$$F_a^1 = \psi_1([\mathcal{R}(F_v); z]), \tag{1}$$

where $z$ is a noise drawn from a standard normal distribution, $\mathcal{R} : \mathbb{R}^{T \times D} \to \mathbb{R}^{\frac{F}{4} \times T \times D}$ is a repeat operator that forms a 3D tensor of height $\frac{F}{4}$, width $T$, and channel $D$ by repeating the input feature $\frac{F}{4}$ times, and $[\,;\,]$ represents concatenation.

The objective of the subsequent generators is to refine the coarse speech representation by jointly modelling the local visual features and global visual context. To this end, a visual context attention module, composed of a global visual encoder and audio-visual attentions, is proposed. The global visual encoder $\phi_c$ derives the global visual features $C_v \in \mathbb{R}^{T \times D}$ by considering the relationships of the entire local visual features $F_v$ through bi-RNN. Then, the audio-visual attention finds the complementary cues (*i.e.*, global visual context) for the speech synthesis, considering the importance of global visual features $C_v$ accordingly with the speech representation $F_a^i \in \mathbb{R}^{F_i \times T_i \times D_i}$ of $i$-th resolution. The audio-visual attention can be denoted as following equations,

$$Q_i = \mathcal{F}(F_a^i) W_q^i, \quad K_i = C_v W_k^i, \quad V_i = C_v W_v^i,$$
$$F_c^i = \mathcal{S}(softmax(\frac{Q_i K_i^\top}{\sqrt{d}}) V_i), \tag{2}$$

where $F_c^i$ represents the global visual context obtained, $W_q^i \in \mathbb{R}^{F_i D_i \times d}$, $W_k^i \in \mathbb{R}^{D \times d}$, and $W_v^i \in \mathbb{R}^{D \times \frac{F_i D_i}{\alpha}}$ represent query, key, and value embedding weights, respectively, $\mathcal{F} : \mathbb{R}^{F_i \times T_i \times D_i} \to \mathbb{R}^{T_i \times F_i D_i}$ is a flatten operator that merges the spectral dimension and channel dimension of speech representation, $\mathcal{S} : \mathbb{R}^{T_i \times \frac{F_i D_i}{\alpha}} \to \mathbb{R}^{F_i \times T_i \times \frac{D_i}{\alpha}}$ is a split operator that maps the 2D tensor into 3D tensor, and $\alpha$ is dimensionality reduction ratio. Note that the audio-visual attention is based on the scaled dot product attention [29] with multi-modal inputs. The visual context attention module and the audio-visual attention are illustrated in Fig.2.

The obtained global visual context $F_c^i$ is concatenated with the coarse speech representation $F_a^i$, and the subsequent generators refine the speech representation iteratively with the following equation,

$$F_a^{i+1} = \psi_i([F_a^i; F_c^i]), \text{ for } i = 1, 2, \ldots, n-1. \tag{3}$$

Therefore, the following generators can get hints for synthesizing an accurate speech from the global visual context during finding the viseme-to-phoneme mapping using local visual features.

In addition, to generate an image (*i.e.*, mel-spectrogram) with fine details, multi-discriminators are utilized following [25]. The multi-scale mel-spectrograms $(\hat{y}_1, \hat{y}_2, \ldots, \hat{y}_n)$ are synthesized from each generated speech representation $(F_a^1, F_a^2, \ldots, F_a^n)$ through $1 \times 1$ convolutions and passed into the multi-discriminators, as illustrated in Fig.1.

## 3.2 Synchronization

Synthesizing a synchronized speech with an input lip movement video is important since human is sensitive to the audio-visual misalignment. In order to generate a synchronized speech, we exploit two strategies: 1) synthesizing the speech by maintaining the temporal representations of input video, and 2) providing a guidance to the generator to focus on the synchronization.

As mentioned above, each visual representation $f_v^t$ encoded from $\phi_v$ contains local-level lip movement information. We design the generator to synthesize the speech conditioned on the local visual features $F_v$ without disturbing its temporal information, so that the output speech can be naturally synchronized through the mapping of viseme-to-phoneme.

Furthermore, to guarantee the synchronization, we adopt a modern deep synchronization concept [35] that learns not only synced audio-visual representations but also discriminative representations in a self-supervised manner. To this end, a local audio encoder $\phi_a$ is introduced that encodes local audio features, $F_a = \{f_a^1, f_a^2, \ldots, f_a^T\} \in \mathbb{R}^{T \times D}$, from the ground-truth mel-spectrogram $y$. With the encoded local audio features $F_a$ and the local visual features $F_v$, a contrastive learning is performed to learn the synchronized representation by using the following InfoNCE loss [36, 37],

$$\mathcal{L}_c(F_a, F_v) = -\mathbb{E}\left[\log\left(\frac{\exp(r(f_a^j, f_v^j)/\tau)}{\sum_n \exp(r(f_a^j, f_v^n)/\tau)}\right)\right],\tag{4}$$

where $r$ represents the cosine similarity metric, $\tau$ is temperature parameter, and $f_a$ and $f_v$ share the same temporal range. The loss function guides to assign high similarity to aligned pairs of audio-visual representations and low similarity to misaligned pairs. We can obtain the synchronization loss for the two encoders $\mathcal{L}_{e\_sync} = \frac{1}{2}(\mathcal{L}_c(F_a, F_v) + \mathcal{L}_c(F_v, F_a))$, where the second term is formed in a symmetric way of Eq.(4) with negative audio samples. In addition, the audio features $\hat{F}_a^n$ encoded from the last generated mel-spectrogram $\hat{y}_n$ is also compared with the visual representations to guide the generator to synthesize the synchronized speech. It is guided with the loss function, $\mathcal{L}_{g\_sync} = ||1 - r(\hat{F}_a^n, F_v)||_1$, which maximizes the cosine similarity between the generated audio features and the given visual features leading the generated mel-spectrogram to be synchronized with the input video. Finally, the final loss for the synchronization is defined as $\mathcal{L}_{sync} = \mathcal{L}_{e\_sync} + \mathcal{L}_{g\_sync}$.

## 3.3 Loss functions

To generate realistic mel-spectrogram, the objective function for the generator parts of the VCA-GAN is defined as

$$\mathcal{L} = \mathcal{L}_g + \lambda_{recon}\mathcal{L}_{recon} + \lambda_{sync}\mathcal{L}_{sync},\tag{5}$$

where $\lambda_{recon}$ and $\lambda_{sync}$ are the balancing weights. $\mathcal{L}_g$ represents GAN loss that jointly models the conditional and unconditional distributions as follows,

$$\mathcal{L}_g = -\frac{1}{2}\mathbb{E}_i[\log D_i(\hat{y}_i) + \log D_i(\hat{y}_i, \mathcal{M}(C_v))],\tag{6}$$

where $D_i$ represents the $i$-th discriminator. The first term is unconditional GAN loss that makes the generated mel-spectrogram to be real, and the second term is conditional GAN loss that guides the generated mel-spectrogram should match with the abbreviated global visual context $\mathcal{M}(C_v)$, where $\mathcal{M}(\cdot)$ represents temporal average pooling.

To complete the GAN training, the discriminator loss is defined as

$$\mathcal{L}_d = -\frac{1}{2}\mathbb{E}_i[\log D_i(y_i) + \log(1 - D_i(\hat{y}_i)) + \log D_i(y_i, \mathcal{M}(C_v)) + \log(1 - D(\hat{y}_i, \mathcal{M}(C_v)))].\tag{7}$$

Finally, the reconstruction loss $\mathcal{L}_{recon}$ is defined with the following L1 distance between generated and ground truth mel-spectrograms of $i$-th resolution,

$$\mathcal{L}_{recon} = \mathbb{E}_i[||y_i - \hat{y}_i||_1].\tag{8}$$

### 3.4 Waveform conversion

The generated mel-spectrogram can be directly utilized for diverse applications, such as Automatic Speech Recognition (ASR) [38, 39], audio-visual speech recognition [40], and speech enhancement [41]. On the other hand, in order to hear the speech sound, the generated mel-spectrogram should be converted into a waveform. It can be achieved by bringing off-the-shelves vocoders [42, 43, 44, 45] that transform the audio spectrogram into waveform. For our experiments, we use Griffin-Lim [46] algorithm. Since the Griffin-Lim algorithm transforms linear spectrogram to waveform, we use additional postnet which learns to map the mel-spectrogram to linear spectrogram similar to [47, 2]. The postnet is trained using L1 reconstruction loss with the ground-truth linear spectrogram.

## 4 Experiments

### 4.1 Dataset

**GRID** corpus [7] dataset is composed of sentences following fixed grammar from 33 speakers. We evaluate our model in three different settings. 1) constrained-speaker setting: subject of 1, 2, 4, and 29 are used for training and evaluation. We follow the dataset split of the prior works [4, 2, 10, 3, 12]. 2) unseen-speaker setting: 15, 8, and 10 subjects are used for training, validation, and test, respectively. The dataset split from [10, 12] is used. 3) multi-speaker setting: all 33 subjects are used both training and evaluation. For the dataset split, we follow the well-known protocol in VSR of [23].

**TCD-TIMIT** dataset [8] is composed of uttering videos from 3 lip speakers and 59 volunteers. Following [4], the data of 3 lip speakers are used for the evaluation in constrained-speaker setting.

**LRW** [9] is a word-level English audio-visual dataset derived from BBC news. It is composed of up to 1,000 training videos for each of 500 words. Since the dataset was collected from the television show, it has a large variety of speakers and poses, presenting challenges on speech synthesis.

### 4.2 Implementation details

For the visual encoder, one 3D convolution layer and ResNet-18 [48], a popular architecture in lip reading [49], are utilized. Three generators are used (*i.e.*, $n$=3) and $2\times$ upsample layer is applied at the last two generators. Each generator is composed of 6, 3, and 3 Residual blocks, respectively. The global visual encoder is designed with 2 layer bi-GRU and one linear layer. For the audio encoder, 2 convolution layers with stride 2 and one Residual block are utilized. The postnet is composed of three 1D Residual blocks and two 1D convolution layers. Finally, the discriminators are basically composed of 2, 3, and 4 Residual blocks. Architectural details can be found in supplementary.

All the audio in the dataset is resampled to 16kHz, high-pass filtered with a 55Hz cutoff frequency, and transformed into mel-spectrogram using 80 mel-filter banks (*i.e.*, $F$=80). For the dataset composed of 25 fps video (*i.e.*, GRID and LRW), the audio is converted into mel-spectrogram by using window size of 640 and hop size of 160. For the 30 fps video (*i.e.*, TCD-TIMIT), the window size of 532 and hop size of 133 are used. Thus, the resulting mel-spectrogram has four times the frame rate of the video. The images are cropped to the center of the lips and resized to the size of $112 \times 112$. During training, the contiguous sequence is randomly sampled with the size of 40 and 50 for GRID and TCD-TIMIT, respectively. During inference, the network generates speech from arbitrary video frame length[1]. For the multi-scale ground-truth mel-spectrograms (*i.e.*, $y_1$ and $y_2$), bilinear interpolation is applied to the ground-truth mel-spectrogram $y$ (*i.e.*, $y_3$). We use Adam optimizer [50] with 0.0001 learning rate. The $\alpha$, $\lambda_{recon}$, and $\lambda_{sync}$ are empirically set to 2, 50, and 0.5, respectively. The temperature parameter $\tau$ is set to 1. For the GAN loss, non-saturating adversarial loss [34] with R1 regularization [51] is used. Titan-RTX is utilized for the computing.

### 4.3 Experimental results

For the evaluation metrics, we use 4 objective metrics: STOI [52], ESTOI [53], PESQ [54], and Word Error Rate (WER). STOI and ESTOI are metrics for measuring the intelligibility of speech audio, and higher scores mean better intelligibility of speech audio. PESQ is a metric for measuring

---

[1]There begins a performance degradation from above about 10 times the length of those used during training.

Table 1: Ablation study in multi-speaker setting on GRID

| Baseline | Proposed Method | | | STOI | ESTOI | PESQ | WER |
| | Visual context attention | Synchro-nization | Multi discriminators | | | | |
|---|---|---|---|---|---|---|---|
| ✓ | ✗ | ✗ | ✗ | 0.726 | 0.584 | 1.917 | 5.43% |
| ✓ | ✓ | ✗ | ✗ | 0.732 | 0.596 | 1.931 | 5.02% |
| ✓ | ✓ | ✓ | ✗ | 0.733 | 0.601 | 1.935 | 4.91% |
| ✓ | ✓ | ✓ | ✓ | **0.736** | **0.604** | **1.961** | **4.80%** |

Table 2: Performance comparison in constrained-speaker setting on GRID

| Method | STOI | ESTOI | PESQ | WER |
|---|---|---|---|---|
| Vid2Speech[1] | 0.491 | 0.335 | 1.734 | 44.92% |
| Ephrat *et al.*[2] | 0.659 | 0.376 | 1.825 | 27.83% |
| Lip2AudSpec[3] | 0.513 | 0.352 | 1.673 | 32.51% |
| 1D GAN-based [10] | 0.564 | 0.361 | 1.684 | 26.64% |
| Lip2Wav [4] | **0.731** | 0.535 | 1.772 | 14.08% |
| VAE-based [11] | 0.724 | 0.540 | 1.932 | - |
| Vocoder-based [12] | 0.648 | 0.455 | 1.900 | 23.33% |
| **VCA-GAN** | 0.724 | **0.609** | **2.008** | **12.25%** |

perceptual quality of speech audio and a higher score implies the better perceptual quality of speech audio. WER measures how correct the predicted text from the generated speech is. A low error rate means better-generated audio containing accurate spoken content. ASR models (modified from [55]) that take the mel-spectrogram as input and are trained on each experimental setting are utilized to measure the WER.

### 4.3.1 Ablation study

We conduct an ablation study in order to confirm the effect of each module in the VCA-GAN. By adopting multi-speaker setting of GRID dataset, we build four variants of the proposed model by gradually adding each proposed module, shown in Table 1. For the WER measurement, a pre-trained ASR model on the same setting of GRID dataset is utilized. The *Baseline* is the model that is trained with $\mathcal{L}_{recon}$ and GAN loss only without the guidance in multi-resolution. It achieves 0.726 STOI, 0.584 ESTOI, and 1.917 PESQ. When the proposed visual context attention module is added, the performances are improved in all the metrics, achieving 0.732, 0.596, 1.931, and 5.02% in STOI, ESTOI, PESQ, and WER, respectively. Please note the performance improvements are the largest when the proposed visual context attention is introduced. This results reflect that jointly modelling the local visual features and the global visual context during the speech synthesis is important for fine speech generation. Further, when the synchronization learning is performed, the intelligibility score, ESTOI, and WER are improved. Finally, by guiding the generation results in multi-resolution with multi-discriminators, we can synthesize the speech sound clearer with the improvement of PESQ.

### 4.3.2 Results in constrained-speaker setting

To compare the proposed VCA-GAN with state-of-the-art methods, we train and evaluate the VCA-GAN in constrained-speaker setting on both GRID and TCD-TIMIT dataset. Table 2 indicates the comparison results on GRID. The proposed VCA-GAN achieves the best performance in all metrics with large margins except STOI, but it shows comparable performance with that of Lip2Wav [4]. In this experiment, WER is measured using Google ASR API for fair comparison, following [4]. The WER measured from the Google API is 12.25% which surpasses the previous state-of-the-art, Lip2Wav, by 1.83%. With the pre-trained ASR, the measured WER of VCA-GAN is 5.83% showing that the generated mel-spectrogram clearly contains the speech content of input lip movements. The comparison results on TCD-TIMIT dataset is shown in Table 3. Similar to the results on GRID dataset, the VCA-GAN shows the state-of-the-art performances in all three metrics. These results confirm that the VCA-GAN consistently synthesizes the speech from the lip movements with high intelligibility and clear sound regardless of the dataset.

Table 3: Performance comparison in constrained-speaker setting on TCD-TIMIT

| Method | STOI | ESTOI | PESQ |
|--------|------|-------|------|
| Vid2Speech[1] | 0.451 | 0.298 | 1.136 |
| Ephrat *et al.*[2] | 0.487 | 0.310 | 1.231 |
| Lip2AudSpec[3] | 0.450 | 0.316 | 1.254 |
| 1D GAN-based [10] | 0.511 | 0.321 | 1.218 |
| Lip2Wav [4] | 0.558 | 0.365 | 1.350 |
| **VCA-GAN** | **0.584** | **0.401** | **1.425** |

Table 4: MOS score comparison with 95% confidence interval computed from the t-distribution

| Method | Intelligibility | Naturalness | Synchronization |
|--------|-----------------|-------------|-----------------|
| 1D GAN-based [10] | $2.74 \pm 0.61$ | $2.28 \pm 0.55$ | $3.69 \pm 0.50$ |
| Lip2Wav [4] | $3.23 \pm 0.48$ | $2.87 \pm 0.54$ | $3.86 \pm 0.38$ |
| Vocoder-based [12] | $3.68 \pm 0.67$ | $3.00 \pm 0.68$ | $4.16 \pm 0.43$ |
| **VCA-GAN** | $\mathbf{4.06 \pm 0.53}$ | $\mathbf{3.35 \pm 0.50}$ | $\mathbf{4.30 \pm 0.37}$ |
| Actual Voice | $4.94 \pm 0.04$ | $4.94 \pm 0.09$ | $4.87 \pm 0.12$ |

Besides the speech quality metrics, we conduct Mean Opinion Score (MOS) tests. The subjects are asked to rate the 1) intelligibility , 2) naturalness, and 3) synchronization of the synthesized speech on a scale of 1 to 5. We ask 12 participants to evaluate samples from 4 different methods and ground-truth ones. The methods are all proceeded with the contrained-setting of GRID dataset and 20 samples per method are rated. As shown in Table 4, the proposed method achieves the best scores on all the three categories, consistent to the results on the above quality metrics. Specifically, the highest synchronization score verifies that the proposed VCA-GAN can generate well-aligned speech with the input lip movements through the proposed synchronization.

### 4.3.3 Results in unseen-speaker setting

Furthermore, we investigate the performance of the VCA-GAN on GRID dataset in unseen-speaker setting following [10, 12], shown in Table 5. We measure the WER using the pre-trained ASR model, trained in unseen-speaker setting of GRID dataset. Compared to the previous works [10, 12], the VCA-GAN outperforms in STOI, ESTOI, and PESQ. Since the model cannot access the voice characteristics of unseen speaker during training, the overall performance is lower than that of the constrained-speaker (*i.e.*, seen-speaker) setting. Even in the challenging setting, the VCA-GAN achieves the best WER score with a large margin compared to the previous methods. The result indicates that the synthesized speech by using VCA-GAN contains the correct content, which is important in unseen-speaker speech synthesis.

### 4.3.4 Results in multi-speaker setting

For verifying the ability of the proposed VCA-GAN on multi-speaker speech synthesis, we train the model by using full data of GRID dataset. Since the voice characteristics varying from different identities, multi-speaker speech synthesis is considered as a challenging problem. Compared to the results of constrained-speaker setting on GRID in Table 2, three metric scores (*i.e.*, STOI, ESTOI, and PESQ) shown in Table 6 do not show significant differences, meaning that the proposed VCA-GAN can synthesize multi-speaker speech without loss of the intelligibility and quality of the speech. Moreover, in Table 6, we compare the Character Error Rate (CER) and WER with a VSR method which predict a text from a given silent talking face video. Even though the VCA-GAN is not trained

Table 5: Performance comparison in unseen-speaker setting on GRID

| Method | STOI | ESTOI | PESQ | WER |
|--------|------|-------|------|-----|
| GAN-based [10] | 0.445 | 0.188 | 1.240 | 38.51% |
| Vocoder-based [12] | 0.537 | 0.227 | 1.230 | 37.97% |
| **VCA-GAN** | **0.569** | **0.336** | **1.372** | **23.45**% |

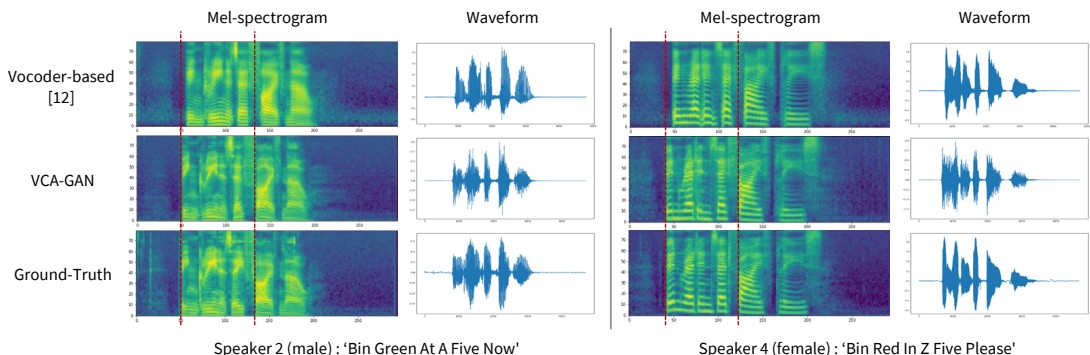

Figure 3: Qualitative results of generated mel-spectrogram and waveform.

Table 6: Performance in multi-speaker setting on GRID

| Method | STOI | ESTOI | PESQ | CER | WER |
|---|---|---|---|---|---|
| LipNet [23] | - | - | - | 2.0% | 5.6% |
| **VCA-GAN** | 0.736 | 0.604 | 1.961 | 1.8% | 4.8% |

with text supervision, the generated speech is intelligible enough to recognize the spoken speech, showing comparable results to the LipNet [23] which is a well-known VSR method.

Finally, we further extend our experiment using LRW dataset, shown in Table 7. While the previous Lip2Wav [4] method utilizes prior knowledge of speakers through speaker embedding [56] for training and inferring on the LRW dataset, the proposed VCA-GAN does not require any prior knowledge of the speakers. Without using additional speaker information, the VCA-GAN outperforms the previous method [4] in three speech quality metrics, STOI, ESTOI, and PESQ. For the WER, Lip2Wav is measured by using Google API while VCA-GAN is measured using the ASR model pre-trained on LRW dataset, so they cannot be directly comparable. However, it clearly shows that the synthesized speech correctly contains the right words even in the challenging environments by outperforming a VSR model [9].

### 4.3.5 Evaluation of audio-visual synchronization

In order to verify how well the generated speech achieves audio-visual synchronization, we adopt two metrics, LSE-D and LSE-C, proposed by [57]. They measure the degree of the synchronization between audio and video using pre-trained SyncNet [35]. The LSE-D measures the feature distance of two modalities, so less value means better synchronization. The LSE-C measures the confidence score, hence the higher score refers to the better audio-video correlation. Table 8 shows the comparison results in two synchronization metrics in constrained-speaker setting on GRID dataset. The proposed VCA-GAN achieves the best score on both LSE-D and LSE-C by surpassing the previous state-of-the-art method [12]. This result shows that the proposed synchronization is effective for synthesizing the in-sync speech with the input lip movement video.

### 4.3.6 Qualitative results

We visualize the generated mel-spectrogram and the waveform of Vocoder-based method[12], the proposed VCA-GAN, and the ground-truth in Fig.3. The results are from the constrained-setting of GRID. We find that the generated mel-spectrograms of VCA-GAN are visually well matched

Table 7: Performance comparison on LRW. † Reported by using Google API.

| Method | STOI | ESTOI | PESQ | WER |
|---|---|---|---|---|
| Chung *et al.*[9] | - | - | - | 38.90% |
| Lip2Wav [4] | 0.543 | 0.344 | 1.197 | †34.20% |
| **VCA-GAN** | **0.565** | **0.364** | **1.337** | 29.95% |

Table 8: LSE-D and LSE-C comparisons for measuring synchronization

| Method | LSE-D ↓ | LSE-C ↑ |
|---|---|---|
| 1D GAN-based [10] | 8.107 | 4.797 |
| Vocoder-based [12] | 6.717 | 6.178 |
| VCA-GAN | **6.698** | **6.373** |

Table 9: Comparison of inference speed with a Seq2Seq-based method.

| Method | Inference time |
|---|---|
| Lip2Wav [4] | 141.73 ms |
| VCA-GAN | 25.89 ms |

with the ground-truth. Moreover, by seeing the red dotted-line on mel-spectrogram that indicates the start of utterance of ground-truth, we can confirm that the results from the VCA-GAN are well synchronized while the results of [12] are slightly shifted to the right, compared to the ground-truth mel-spectrogram.

#### 4.3.7 Computational cost

As the proposed VCA-GAN can synthesize the entire speech with one forward step, we can save the inference time than using a Seq2Seq-based method [4]. In order to examine the improved performance in terms of inference speed, we measure the inference time of a Seq2Seq-based method, Lip2Wav, and VCA-GAN including postnet when generating 3-sec speech. For the computing, Titan RTX is utilized. Table 9 shows the measured inference time of each method. The mean inference time of VCA-GAN for generating 3-sec speech takes 25.89ms and the Lip2Wav needs about 5 times more time than VCA-GAN.

## 5 Limitations and Societal Impacts

This work offers a powerful lip to speech synthesis method. However, as shown in Section 4.3.3, the speech synthesis performances on unseen speakers are still limited compare to the seen speakers. This is because of the unpredictable voice characteristics of unseen speaker that the model cannot properly synthesize. A possible direction for alleviating this limitation is to remove the identity factors of training samples and re-painting them on the generated speech.

With this work, several positive societal benefits can be derived such as assisting human conversations in crowd or silent environments and making it possible to communicate with the speech impaired. In contrast to the advantages, there are also some potential downsides. The lip reading can read the speech by only capturing the lip movement of a certain person, and the visual information can be more easily obtained than the high-quality audio information in a crowded environment or in a long-distance. Thus, the technology is possible of being misused in a surveillance system which can erode individual freedom and damages one's privacy.

## 6 Conclusion

We have proposed a novel Lip2Speech framework, VCA-GAN, which generates the mel-spectrogram using 2D GAN by jointly modelling both local and global visual representations. Specifically, the visual context attention module provides the global visual context to the intermediate layers of the generator, so that the mapping of viseme-to-phoneme can be refined with the context information. Moreover, to guarantee the generated speech to be synchronized with the input lip video, synchronization learning is introduced. Extensive experimental results on three benchmark databases, GRID, TCD-TIMIT, and LRW, show that the proposed VCA-GAN outperforms existing state-of-the-art and effectively synthesizes the speech from multi-speaker.

## Acknowledgments and Disclosure of Funding

We would like to thank the anonymous reviewers for their helpful comments to improve the paper. This work was partially supported by Genesis Lab under a research project (G01210424).

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
