# OpenReview forum: "Lip to Speech Synthesis with Visual Context Attentional GAN"
_NeurIPS.cc/2021/Conference — NeurIPS 2021 Poster_

### Official Review · Reviewer_35i8 · 2021-07-16

**Rating:** 6
**Confidence:** 3

**Summary:**

This paper proposes a lip2speech generative framework based on the proposed Visual Context Attention (VCA) module. The VCA module follows the idea of self-attention and fuses the audio-visual information together to generate finer representations. The proposed pipeline stacks several generators together whose predictions are gradually refined by the VCA module. The global and local clues of input videos are also taken into consideration in this framework. The experiments are conducted on three datasets, while the authors have evaluated their method under different scenarios like constrained-speaker, unseen speaker, etc. The final performance is superior to other baselines.

**Limitations And Societal Impact:**

Yes, I think so.

**Main Review:**

STRENGTH
+ The idea of adding an audio-visual attention module to iteratively refine the representation seems to be novel in the lip2speech field. The motivation for designing the VCA module seems reasonable to me.
+ The performance of the proposed method looks promising. Different scenarios, including constrained-speaker, unseen speaker and multiple-speaker are evaluated, which demonstrates the generality of the proposed pipeline. The ablation study is also performed to illustrate the effectiveness of each component.
+ The code is submitted for reproduction.
+ The paper writing is well organised and easy to follow.

WEAKNESS
- In VCA module, why use speech representation as the query and use visual clues as the key and value?
- For the pipeline design, why choose the number of generators (n) to 3? How will n affect the whole performance?
- It is difficult for me to understand why the Audio-Visual Attention mechanism can generate better audio representations. Can the authors explain this more detailedly and intuitively?
- The performance of audio-visual recognition models on the generated voice can also be a helpful evaluation metric.
- I checked the demo in the supplemental material,  but the results of various methods (author's vs baselines) doesn't seem too much different to me.

Overall, the idea of this paper can be novel in the field of lip2speech, and the motivation and model design looks reasonable. The proposed method is evaluated extensively and the results are superior. I will vote for a marginally above.

**Time Spent Reviewing:**

2

---

> ### Author Response · Authors · 2021-08-10
> **Responses to reviewer 35i8**
>
> **(Q1) Query, Key, and Value selection**
>
> **(A1)**
> In order to define why we choose the speech representation as query and the global visual representations as the key and value, the motivation for introducing the VCA module should be delivered in advance. The main challenge in generating speech from a silent lip video is the existence of homophenes that have similar lip movement while acoustic pronunciations are different. For example, ‘mark’, ‘bark’, and ‘park’ have the same lip movements while the speech sounds are distinct. In order to capture the differences and generating accurate speech from the silent lip video, the context should be considered. Therefore, we built a VCA module that can capture the global context from the entire sequences of input lip video. For providing the global visual context into the generation procedure and refining the coarsely generated speech, we designed the query by using the generated speech features, and key and value by using the visual representations. Thus, the global visual context can be considered for the further speech generation parts.
>
> ---
>
> **(Q2) Number of generators (discriminators)**
>
> **(A2)**
> We would like to thank the reviewer for the valuable comment. In order to show how the number of generators (discriminators) affects the whole performance, we report the speech quality metric performances under the different number of generators (discriminators) shown in Table R4_1.
>
> - **Table R4_1** Speech quality scores of different $n$
> | $n$ | STOI | ESTOI | PESQ |
> |:---:|:---:|:---:|:---:|
> | 1 | 0.733 | 0.601 | 1.935 |
> | 2 | 0.732 | 0.595 | 1.948 |
> | 3 | 0.736 | 0.604 | 1.961 |
>
> By increasing the number of generators (discriminators), the PESQ scores are improved and this result shows that using multi-scale generation is beneficial to synthesize the speech clearer. Therefore, we choose $n$=3 by considering both the performance and model parameters (we did not try more than 3 due to memory constraints).
>
> ---
>
> **(Q3) Why AV attention helps to generate audio**
>
> **(A3)**
> As we have mentioned, the Audio-Visual (AV) attention is constructed to consider the global visual context in the VCA module. In the following, we describe how the AV attention mechanism can help the detailed audio generation. The coarsely generated audio representation at the first generator does not contain any global context since they are synthesized from the local visual representations. Therefore, the coarsely generated speech might not distinguish the homophenes. Through the AV attention, the most suitable global visual representation to the given coarsely generated audio can be searched. The searched global visual representation is obtained as a value in the AV attention and is concatenated with the coarsely generated audio representation. Then, the next generator refines the previous ambiguous speech more elaborately by jointly modelling the global and local representations. Please see Table 1 showing the VCA module improves the performance with the largest margin with the aforementioned role.
>
> ---
>
> **(Q4) Audio-visual recognition models on generated voice**
>
> **(A4)**
> We would like to thank the reviewer for the valuable comment. As the reviewer pointed out, speech recognition using generated speech can be a helpful evaluation metrics to verify the effectiveness of the proposed method. Therefore, we applied a pre-trained Automatic Speech Recognition (ASR) model to the generated speech for measuring intelligibility. The measured WER performance using ASR is reported in Table 2, Table 5, and Table 6. Moreover, we additionally report the WER performance for the ablation study in Table R4_2. Please note that our method is for synthesizing the speech from silent video and thus we used ASR model instead of Audio-Visual Speech Recognition (AVSR) model to measure the intelligibility of the synthesized speech.
>
> - **Table R4_2** WER performances on multi-speaker setting on GRID (will be added in Table1)
> | Baseline | + Visual context attention | + Synchronization | + Multi-discriminators |
> |:------:|:------:|:------:|:------:|
> | 5.434% | 5.015% | 4.908% | 4.797% |
>
> ---
>
> **(Q5) About results shown in demo**
>
> **(A5)**
> We would like to clarify the performance comparisons shown in the demo. First, the 1D GAN-based method [R10] shows speech with a rather noisy background, the Lip2Wav [R4] has some faults in speaking correct words, and the Vocoder-based method [R12] shows a rather unnatural voice (robotic). Please see the Mean Opinion Score (MOS) comparison results shown in Table 4 which performed with more generated samples.
>
> ---
>
> [R#] represents reference # in the manuscript.

---

### Official Review · Reviewer_uxUX · 2021-07-17

**Rating:** 7
**Confidence:** 4

**Summary:**

In this work, the authors study the problem of synthesizing speech corresponding to a silent video. The main novelties proposed in this work are: a proposal to incorporate a “global visual context” which is hypothesized to help the model resolve ambiguity in the viseme-to-phoneme mapping; an explicit synchronization technique to ensure that the generated speech is time-aligned with the video; and the use of multiple discriminators to which allow the iterative generation of more fine-grained speech representations. Essentially, the authors propose to model the task as video to image translation, where the video corresponds to a log-mel spectrogram representation of the target speech.


**Limitations And Societal Impact:**

One possible concern is that this work could be used for surveillance when only a video but no audio is available. The authors mention that this is a potential concern.

**Main Review:**


This is a very well written paper (although it does have a number of minor typographical and grammatical errors; few are listed below) which is generally well motivated and easy to follow. The experimental section, in particular, is comprehensive and demonstrates that the proposed system outperforms previous works with larger gains in unseen and multi-speaker settings (which are more challenging to begin with).

I had a few minor comments which if resolved would strengthen the paper further.

- **Ablation results in Table 1 and Table 3**: It would be good to include WER numbers as well for the ablation study in Table 1 and the results in Table 3 to show fidelity in addition to intelligibility and quality, since it is also useful to see how well the reconstructed speech matches the ground-truth audio.

- **Captured global context**: A point that is made in the paper, and in Table 1, is that the use of the global context helps resolve ambiguities. It would be interesting to present the results of some error analysis to show how the global context helps improve performance (WER results in Table 1, would help clarify this somewhat as well).

- **Length of test videos**: In Section 4.2, the authors mention that during training the videos are segmented to smaller chunks, but during inference the authors evaluate the full videos. I was curious if the authors have performed any experiments to study the overall performance as a function of video length. Are the global features still useful for very long videos, for example? It would be useful to mention some statistics about the test video lengths for the various datasets (perhaps in a footnote if space is limited).

- **Intelligibility and Quality Metrics**: Minor comment: In section 4.3, it might be useful to mention a couple of sentences about the STOI/ESTOI/PESQ metrics to help the reader understand what these capture, whether higher/lower are better and what level of differences are generally significant.

Typographical and Grammatical Errors:
1. Page 3, Line 98: “... contexture design … ”
2. Page 3, Section 3: “... as the input silence video.” --> “... as the input silent video.”
3. Page 3, Section 3: “... is designed to have 4 times longer ...” -->  “... is designed to be 4 times longer ...”
4. Page 3, Section 3: “... dealing the mel-spectrogram … ” -->  “... treating the mel-spectrogram ...”
5. Eqn 6: It might be better to use the notation L_{g}^{i} instead of L_g
6. Page 6, Line 221: “... four times the frame of the ...” --> “... four times the frame rate of the ...”
7. Page 8, Line 287: “Even the VCA-GAN is ...” --> “Even though the VCA-GAN is ...”


**Time Spent Reviewing:**

6

---

> ### Author Response · Authors · 2021-08-10
> **Responses to reviewer uxUX**
>
> **(Q1) WER results on Table1 and Table3**
>
> **(A1)**
> We would like thank to the reviewer for the valuable comment. We will add the WER performances in Table 1 to show how well the reconstructed speech matches the ground-truth audio (shown in Table R3_1). For Table 3, we report the WER performances obtained using Google API below in Table R3_2, but we would like to clarify why we did not include them in the original manuscript. TCD-TIMIT database [R8] is originally proposed to analyze the relation of phonemes and visemes using the paired audio-visual corpus. Therefore, the utterances are designed to cover overall phonemes and visemes. This is the reason that many previous works that utilized the TCD-TIMIT database report viseme-level or phoneme-level performance (PER) [R8, 1, 2, 3, 4, 5] instead of using word-level performance (WER) that is barely used for the dataset. Moreover, the dataset is composed of small training samples (377 for each Lip speaker) to train the ASR model compared to GRID (1,000 for each speaker) while containing more words. This leads not enough performance (42% WER / 13% CER) of the ASR model trained with TCD-TIMIT to evaluate the generated speech.
>
> - **Table R3_1** WER performances on multi-speaker setting on GRID (will be added in Table1)
> | Baseline | + Visual context attention | + Synchronization | + Multi-discriminators |
> |:------:|:------:|:------:|:------:|
> | 5.434% | 5.015% | 4.908% | 4.797% |
>
> - **Table R3_2** WER performances on constrained-speaker setting on TCD-TIMIT
> | Vid2Speech [R1] | Ephrat et al. [R2] | Lip2AudSpec [R3] | 1D GAN-based [R10] | Lip2Wav [R4] | **VCA-GAN** |
> |:---:|:---:|:---:|:---:|:---:|:---:|
> | 75.52% | 61.86% | 49.13% | 53.52% | 31.26% | 49.09% |
>
> ---
>
> **(Q2) The effect of capturing global context**
>
> **(A2)**
> In order to confirm the effectiveness of capturing global context on generating speech, we added WER results in Table R3_1. The improved WER performance implies the synthesized speech contains clearer wordings. Moreover, please note that the performance improvement in Table 1 is the largest when the proposed visual context attention is applied.
>
> ---
>
> **(Q3) Length of test videos**
>
> **(A3)**
> The test videos of GRID have a fixed frame length of 75 and are used to train with randomly sampled 40 frames. We experimentally found that the training with sampled frames (i.e., 40) instead of the entire frames (i.e., 75 frames) is beneficial to the final performance. This could be the reason for the data augmentation effects. Moreover, please find the attention map shown in Figure 1 in the supplementary. It shows the audio-visual attention map during the test (i.e., 75 frames). We can find that the audio-visual attention well attends to the global visual feature and its neighborhoods in accordance with the corresponding audio frame. This is because of the shift-invariant property of convolution and the effectiveness of scaled dot product attention on sequence modelling that is proven in the natural language processing area [R27].
> In order to analyze the effect of training with smaller chunks on the inference phase that using full video, we evaluate the model trained with 40 frames using different video lengths (75 frames to 750 frames) in the inference phase. The videos longer than 75 frames are obtained by repeating the original inference video. And we find that there is minor performance degradation in the three speech quality metrics (STOI, ESTOI, and PESQ) for very long videos as shown in Table R3_3.
> Therefore, in practice, our model is valid to accept up to 10 times the length of those used during training to expect inference of good quality.
>
> - **Table R3_3** Speech quality scores of different inference frame lengths
> | Frame Length | STOI | ESTOI | PESQ |
> |:---:|:---:|:---:|:---:|
> |75|0.755|0.612|2.167|
> |375|0.752|0.609|2.143|
> |750|0.755|0.605|2.132|
>
> ---
>
> **(Q4) Intelligibility and Quality Metrics**
>
> **(A4)**
> As per the reviewers’ suggestion, we add the description for three speech quality metrics as follows,
>
> STOI(Short Term Objective Intelligibility) and ESTOI(Extended Short Term Objective Intelligibility) are metrics for measuring the intelligibility of speech audio, and higher scores mean better intelligibility of speech audio. PESQ(Perceptual Evaluation of Speech Quality) is a metric for measuring the perceptual quality of speech audio. The score is varying from -0.5 to 4.5 and a higher score means the better perceptual quality of speech audio. WER(Word Error Rate) measures how correct the predicted text from the generated speech is, using an ASR model. A low error rate implies better-generated audio in both intelligibility and quality.
>
> ---
>
> **(Q5) Typographical and Grammatical errors**
>
> **(A5)**
> We would like to thank the reviewer for the valuable comment. We will proofread the manuscript and correct English language mistakes.
>
> ---
>
> [R#] represents reference # in the manuscript.
>
> **References**
>
> - [1] Chen, Weicong, et al. "DualLip: A System for Joint Lip Reading and Generation." Proceedings of the 28th ACM International Conference on Multimedia. 2020.
> - [2] Abdelaziz, Ahmed Hussen. "Turbo Decoders for Audio-Visual Continuous Speech Recognition." INTERSPEECH. 2017.
> - [3] Zhang, Shiliang, et al. "Robust audio-visual speech recognition using bimodal DFSMN with multi-condition training and dropout regularization." ICASSP 2019-2019 IEEE International Conference on Acoustics, Speech and Signal Processing (ICASSP). IEEE, 2019.
> - [4] Abdelaziz, Ahmed Hussen. "Comparing fusion models for DNN-based audiovisual continuous speech recognition." IEEE/ACM Transactions on Audio, Speech, and Language Processing 26.3 (2017): 475-484.
> - [5] Li, Wei, et al. "Improving audio-visual speech recognition performance with cross-modal student-teacher training." ICASSP 2019-2019 IEEE International Conference on Acoustics, Speech and Signal Processing (ICASSP). IEEE, 2019.

---

> > ### Comment · Reviewer_uxUX · 2021-08-24
> > **Thanks for your replies to the review comments**
> >
> > Thanks for responding to the review comments and for clarifying the issues raised in the review. I would like to retain my original score.

---

### Official Review · Reviewer_Nvn1 · 2021-07-18

**Rating:** 6
**Confidence:** 4

**Summary:**

This paper describes a synthesis approach that uses visual lip movement as input and generates speech as output.  They use a GAN style synthesizer. The most novel contribution is the synchronization technique to align visual and speech signals.

**Ethical Concerns:**

The Ethics Guidelines includes an example that covers this submission and similar work

5. Develop or extend harmful forms of surveillance. For example: could it be used to collect or analyze bulk surveillance data to predict immigration status or other protected categories, or be used in any kind of criminal profiling?

Visual information is much easier to collect at scale than high quality audio information.  E.g. in a crowded public place recording audio of a single speaker is very challenging, while collecting visual imaging of a single person is fairly straightforward.  Visual speech recognition and speech synthesis would extend the ability to surveil people's speech in areas where audio collection is impossible, but visual collection is possible.


**Ethics Review Area:**

["Inappropriate Potential Applications & Impact  (e.g., human rights concerns)"]

**Limitations And Societal Impact:**

The paper does include a section on Limitations and Societal Impact. While the potential abuse of visual based speech synthesis or recognition is acknowledged, there is no mention of mitigation factors or discussion of the balance between positive and negative societal impact.

**Main Review:**

This paper is about GAN synthesis of speech based on lip images.  This is not the first paper to have done this: e.g. "Video-Driven Speech Reconstruction using Generative Adversarial Networks" Vougioukas1 et al. 2019 https://arxiv.org/pdf/1906.06301.pdf The architectural modification is the use of a hierarchical structure using local and global features.


"However, only watching short clip-level (i.e., local) lip movements could be challenging to distinguish the homophones." Define “homophones” here?  The most common usage of "homophones" is to refer to words with different written forms but identical spoken forms, like "their" and "there". it is not clear how global level information is helpful in distinguishing these.

Line 111: "...global visual context, can provide additional information that alleviates the ambiguity of homophones"	Please explain how, perhaps with an example?

Line 171 ", which maximizes the cosine similarity between the generated audio features and the given visual features leading  the generated mel-spectrogram to be synced with the input video.” In order to perform the synchronization between visual inputs and audio outputs, the two features mush share a common representation space.  Does this choice improve or impair quality?  	Per Table 1 synchronization gives a very minor win, while the largest improvement comes from the use of visual attention.

Table 2: The order of systems in Table2 is unclear.  Why this sorting?

Table 1: It would be useful to compare the pretrained ASR (5.83%) to any other system.  Without that comparison, this number isn’t meaningful, and probably should be removed.

Line 294: “In addition, for the WER, Lip2Wav is measured by using Google API while VCA-GAN is measured using the ASR model pre-trained on LRW dataset, so they cannot be directly comparable. . However, it clearly shows that the synthesized speech correctly contains the right words even in the challenging environments.”  It is impossible to draw comparisons here, this mismatched evaluation of intelligibility should be removed.  It would be better to either evaluate both using the same system, or leave the comparison out.

It would be helpful to have some subjective measure of the value of synchronized images and audio.





**Needs Ethics Review:**

Yes

**Time Spent Reviewing:**

2

---

> ### Author Response · Authors · 2021-08-10
> **Responses to reviewer Nvn1**
>
> **(Q1) About GAN-based prior work**
>
> **(A1)**
> We would like to clarify that the GAN-based prior work “Video-Driven Speech Reconstruction using Generative Adversarial Work” [R10] is different from our work. Even though they used GAN to generate speech, their GAN model is a 1D-based model that directly predicts the speech in a raw waveform. However, it is well-known that generating raw waveform directly is difficult due to the lack of suitable loss functions (e.g., L1 loss). In contrast, we consider the spectrogram as an image that represents time and spectral features with its width and height respectively, so we adopted a 2D GAN technique with a reconstruction loss function. Therefore, we redesigned not only the hierarchical architecture but also the whole generation framework composed of 2D image generation with explicit modelling of global and local representations, which is distinct to the 1D GAN-based prior work [R10].
>
> ---
>
> **(Q2) About homophenes & Global visual context in alleviating the ambiguity of homophenes**
>
> **(A2)**
> We would like to thank the reviewer for the valuable comment. Firstly, we would like to clarify that the word “homophone” used in the manuscript is intended to mean a set of words that shows the same lip movements (i.e., visemes) with different pronunciations (i.e., phonemes) rather than different written forms but having identical spoken forms. We note that the correct word for describing this is "homophene" not "homophone" and is used without clear distinction in previous works (homophone in [R14], homophene in [R4]). We will modify the word “homophone” used in the manuscript to “homophene” for a clear description.
>
> Such homophenes include ‘m’, ‘b’, and ’p’ (e.g., mark, bark, and park) and they are cannot be distinguished by watching the lip movement only. In order to correctly distinguish the homophenes, the context should be considered (e.g., go to a mark for rest vs. go to a bark for rest vs. go to a park for rest). This is the motivation of our work that previous works which predict the auditory feature from a short video clip could fail on distinguishing the homophenes. In the proposed method, the global-level information capturing the context can provide additional information besides the local-level information so that the generator can synthesize the speech by distinguishing the homophenes.
>
> ---
>
> **(Q3) Choice of synchronization**
>
> **(A3)**
> The choice of using cosine similarity and learning a shared latent space of visual and audio modalities are motivated by the modern success in audio-video synchronization research [R32]. In [R32], the authors showed that contrastive learning using audio-video data is effective in learning synchronized audio-video representations. Moreover, they showed that contrastive learning (i.e., learning shared latent space) is beneficial to the lip-reading task by obtaining more discriminative representations in a self-supervised manner. Therefore, we adopted their idea to synthesize synchronized audio representation with input video and detailed speech.
> To verify that this choice improves the quality of generated speech, we additionally report WER performances that reflect the clarity of the generated speech in Table R2_1. The WER is measured by using a pre-trained ASR model trained on the same multi-speaker setting on GRID. By using the synchronization, the WER improves about 0.1%. This improvement results from our training strategy where the model learns discriminative representation through contrastive learning.
>
> - **Table R2_1** WER performances on multi-speaker setting on GRID (will be added in Table1)
> | Baseline | + Visual context attention | + Synchronization | + Multi-discriminators |
> |:------:|:------:|:------:|:------:|
> | 5.434% | 5.015% | 4.908% | 4.797% |
>
> Moreover, the red-boxed region of mel-spectrogram in the third column of Fig.2 in the supplemental material shows that the blurry synthesized region becomes clearer with the synchronization learning.
>
> ---
>
> **(Q4) Table 2 sorting**
>
> **(A4)**
> We sorted the methods by published date. Thus, the lower placed method is the more recent method.
>
> ---
>
> **(Q5) Table 2 ASR performance**
>
> **(A5)**
> As per the reviewer’s comment, we will remove the WER performance measured by pre-trained ASR from Table 2.
>
> ---
>
> **(Q6) WER comparison on LRW dataset**
>
> **(A6)**
> As per the reviewer’s comment, we will leave the comparison out for the WER performance on LRW dataset in the manuscript.
>
> ---
>
> **(Q7) Subjective measurement of the value for synchronization**
>
> **(A7)**
> For evaluating the synchronization, we used two metrics for measuring the synchronization. First is the human subjective evaluation (MOS test) shown in Table 4. The second is using metrics proposed to measure the synchronization, LSE-C and LSE-D, shown in Table 9 in the supplementary. Please note that the baseline model without the proposed synchronization achieves 7.055 LSE-D (lower is better) and 6.193 LSE-C (higher is better). The 1D GAN-based method [R10] achieves 8.107 LSE-D and 4.797 LSE-C, the Vocoder-based method [R12] achieves 6.717 LSE-D and 6.178 LSE-C, and the proposed VCA-GAN achieves 6.698 LSE-D and 6.373 LSE-C. From the two results, we can confirm the synthesized speech from the proposed method is well synchronized with the input lip video.
>
> ---
>
> **Ethical concerns**
>
> As the reviewer pointed out, as many beneficial techniques also have their negative social effects, the Lip2Speech or Lip-reading technologies might be misused for harmful forms of surveillance. However, we expect the concern for the negative social effects of the technology to be resolved if the technique is made use of within the border of appropriate legal systems. For example, permission to use the technology in a hospital is very beneficial for conversation with the patients who cannot make voice, and it is already under trial runs in the UK (https://www.sravi.ai/). Another example, permission to use the technology for police can prevent crimes or boost criminal investigation. Therefore, in order to highlight the positive societal effects while preventing the side effects, the development of Lip2Speech or Lip-reading technologies should be processed along with the legal systems.
>
> ---
>
> [R#] represents the reference # in the manuscript.

---

> > ### Comment · Reviewer_Nvn1 · 2021-09-07
> > **Thank you**
> >
> > Thank you for this response. I appreciate the effort in addressing the comments raised in my and other reviews.

---

### Official Review · Reviewer_jtpu · 2021-07-18

**Rating:** 6
**Confidence:** 4

**Summary:**

The paper presents a visual context attentional GAN for synthesising speech from silent videos. It generates the mel-spectrogram by jointly modelling both local and global visual representations; furthermore a synchronisation technique is proposed. The proposed method is evaluated on three commonly used datasets and shows superior performance as compared with state-of-the-art methods in the literature.

**Limitations And Societal Impact:**

Yes

**Main Review:**

Table 1 presents an ablation study in multi-speaker setting on GRID. It is unclear what the difference between the last row of Table 1 and the last row of Table 6, both representing VCA-GAN but with minor performance difference - good to clarify. While the ablation study shows each component contributes to the performance improvement, but the improvement brought by each component is rather small. More specifically, all three components together improve STOI from 0.726 to 0.736 and PESQ from 1.917 to 1.961. This study makes it less clear why the proposed system works well on all three benchmark data sets.

Good to synchronise audio and visual information. The three metrics used in the paper, also commonly used in the literature, are unable to show its whole merit since they are audio only metrics. Watching the demo video in supplementary material does not show the audio and visual sequences are synchronised well though.


**Time Spent Reviewing:**

4

---

> ### Author Response · Authors · 2021-08-10
> **Responses to reviewer jtpu**
>
> **(Q1) The performance difference between Table 1 and Table 6**
>
> **(A1)**
> We thank the reviewer for the valuable comment. As the reviewer pointed out, the reported performances for the three speech quality metrics of Table 6 are mistyped. The performances of VCA-GAN in Table 6 should be revised to 0.736, 0.604, 1.961, 1.8%, and 4.8%, for STOI, ESTOI, PESQ, CER, and WER, respectively.
>
> ---
>
> **(Q2) Rather small improvement by each component**
>
> **(A2)**
> Firstly, we kindly point out that the performance improvement of 0.01 in STOI and 0.05 in PESQ are not so marginal. Please note that the recent state-of-the-art methods [R4, R11, R12] (published in 2020-2021) are competing within 0.007 STOI and 0.03 PESQ scores as shown in Table 2. Moreover, we additionally report WER performances for the ablation study shown in Table R1_1, and the results show about 0.6% improvement by applying all three components together.
>
> - **Table R1_1** WER performances on multi-speaker setting on GRID (will be added in Table1)
> | Baseline | + Visual context attention | + Synchronization | + Multi-discriminators |
> |:------:|:------:|:------:|:------:|
> | 5.434% | 5.015% | 4.908% | 4.797% |
>
> Secondly, we would like to clarify that the “Baseline” in Table 1 results from one of our contributions (Contribution 2 in Introduction). To the best of our knowledge, there is no work that predicts the spectrogram from silent lip video by adopting the image generation technique. In this paper, we synthesized the mel-spectrogram as an image using 2D GAN. Since the value of mel-spectrogram is smooth and it gradually changes over time compared to the raw waveform format, adopting the image generation technique is effective for predicting the spoken speech from silent lip movements. Therefore, our “Baseline” model, one of our contributions, basically achieves high performance in all three speech quality metrics. In order to verify that our “Baseline” model is comparable with previous methods, we report the performance of “Baseline” in constrained speaker setting on GRID dataset (STOI: 0.7196, ESTOI: 0.5984, PESQ: 1.9602). In addition to the “Baseline” architecture, we improve the performance by proposing the visual context attention module, synchronization learning, and multi-scale generation. The ablation study shown in Table 1 verifies that each proposed method can further improve the speech synthesis performance.
>
> ---
>
> **(Q3) About metrics for synchronization**
>
> **(A3)**
> For measuring synchronization, we utilized two different metrics. The first is human subjective evaluation shown in Table 4. The second is quantitative results shown in Table 9 in supplementary using LSE-D (higher is better) and LSE-C (lower is better) that measure the degree of synchronization of audio and video. Please note that the baseline model without synchronization learning achieves 7.055 LSE-D and 6.193 LSE-C. The 1D GAN-based method [R10] achieves 8.107 LSE-D and 4.797 LSE-C, the Vocoder-based method [R12] achieves 6.717 LSE-D and 6.178 LSE-C, and the proposed VCA-GAN achieves **6.698** LSE-D and **6.373** LSE-C. The two results confirm that the proposed method is effective in synthesizing synchronized speech from the silent lip video.
>
> ---
>
> [R#] represents the reference # in the manuscript.

---

### Review · Ethics_Reviewer_8nNQ · 2021-07-30

**Recommendation:**

The concern that the proposed technology could be used for surveillance is valid. I recall that similar concerns were raised when researchers at MIT were interested in designing "visual microphones" ([see e.g., this PhD thesis](http://abedavis.com/files/papers/thesis.pdf)). I would recommend that the authors add a meaningful discussion of this possibility that is broadly accessible that, and that cites this work. The goal here is to be transparent and to acknowledge the risks.

**Ethical Issues:**

Yes

**Ethics Review:**

The ethics review was initiated based on a comment of using the technology for the purposes of surveillance. Specifically: "Visual information is much easier to collect at scale than high-quality audio information. E.g. in a crowded public place recording audio of a single speaker is very challenging, while collecting visual imaging of a single person is fairly straightforward. Visual speech recognition and speech synthesis would extend the ability to surveil people's speech in areas where audio collection is impossible, but visual collection is possible."

---

> ### Author Response · Authors · 2021-08-19
> **Response to ethics reviewer 8nNQ**
>
> We would like to thank the reviewer for the valuable comment. As per the reviewer’s comment, we add the discussion of the possibility of abusing the lip-reading technology (output form of either text or speech). The lip-reading can read the speech by only capturing the lip movement of a certain person, and the visual information can be more easily obtained than the high-quality audio information in a crowded environment or in a long-distance. Thus, the technology is possible of being misused in a surveillance system which can erode individual freedom and damages one’s privacy. Therefore, researchers in this area should acknowledge the risks when developing their technology. In order to reduce the possibility of abuse, researchers can be more careful of distributing the source code of their lip-reading technology. Rather than making the code publicly available, the researchers should keep on track whom the code is provided to (i.e., gated release of the model). Moreover, society (not only the authors) also needs to promote such technical development by establishing appropriate legal systems to which the technology can be applied for positive purposes. Besides the possible negative effects, we highlight that there are also many positive societal effects. The technology can be applied in 1) video conferencing in a silent or crowded environment, 2) audio enhancement using visual information, 3) conversation in a long-distance, and 4) conversation with people who cannot make a voice.

---

### Review · Ethics_Reviewer_pVfn · 2021-08-09

**Recommendation:**

Given that there is a concerning surveillance application for this type of method, the authors should do more to describe the potential for abuse here, including recommendations for how researchers in this space can responsibly conduct their work (choice of datasets, protocol for sharing models, etc.). Some discussion on task definition and data modality would also be relevant. For example, the observation (due to Reviewer Nvn1) that video may be easy for a surveillance organization to collect (compared with audio) seems worth mentioning. On the other hand, it seems that an AI offering video-to-text-transcript conversion might be preferable for surveillance compared with the type of video-to-speech AI discussed here, as text can be stored and indexed more efficiently.

**Ethical Issues:**

Yes

**Ethics Review:**

This submission was flagged for ethics review by Reviewer Nvn1, who correctly identified that this paper could potentially be misused for surveillance purposes. The authors have also briefly mentioned this issue (line 315). As both the authors and reviewer acknowledged, this critique is not specific to the proposed method, but instead applies to all Lip2Speech approaches. However, my feeling is that the authors should provide a deeper discussion on this topic.

---

> ### Author Response · Authors · 2021-08-19
> **Response to ethics reviewer pVfn**
>
> We would like to thank the reviewer for the valuable comment. As the Reviewer Nvn1 and the Ethics Reviewers pointed out, the lip-reading technology (either in the form of text or speech) has the potential for abuse of surveillance applications such as an individuals’ illegal profiling which leads to invasion of privacy. This is possible because visual information of a certain person can be obtained more easily than audio information from a far distance or in a crowded situation. Nevertheless, the lip-reading technology can be utilized in a large range of beneficial applications such as 1) video conferencing in a silent or crowded environment, 2) audio enhancement using visual information, 3) conversation in a long-distance, and 4) conversation with people who cannot make a voice. Therefore, the researchers in this area including us should take into account the potential abuse while researching with positive intend. To minimize the potential negative societal effect, the researchers can be more careful of distributing the source code of their lip-reading technology. Rather than making the code publicly available, the researchers should keep on track whom the code is provided to (i.e., gated release of the model). Furthermore, society (not only the authors) also needs to promote such technical development by establishing appropriate legal systems to which the technology can be applied for positive purposes.

---

### Decision · Program_Chairs · 2021-09-27

**Decision:**

Accept (Poster)

**Comment:**

The authors investigate speech generation out of silent videos based on the so-called visual context attentional GAN.  The problem is interesting and challenging.  The authors propose to use global visual context to reduce the ambiguity when mapping visemes to phonemes and introduce a synchronization mechanism via contrastive learning to make speech and lip movements in sync.  Extensive experiments and evaluation are conducted to compare its performance with a variety of existing techniques.  The proposed VCA-GAN reports the state-of-the-art performance in the lip-to-speech domain.  The paper is well written and easy to follow.  The experiments are extensive yet controlled.  The rebuttal has clarifies most of the concerns in the review.  All reviewers are supportive on accepting the paper.  The authors should revise the submission to address the concerns (including the ethical concerns) raised by the reviewers.